# Dental Criteria Could Alert for Malnutrition Risk and Inappropriate Choice of Food Texture in Older Subjects with Dementia: An Analytical Cross-Sectional Study

**DOI:** 10.3390/ijerph192215086

**Published:** 2022-11-16

**Authors:** Nada El Osta, Amine Wehbe, Nelly Sleiman, Noemie Drancourt, Lana El Osta, Martine Hennequin

**Affiliations:** 1Centre de Recherche en Odontologie Clinique (CROC), Université Clermont Auvergne, 63000 Clermont-Ferrand, France; 2Faculty of Medicine, Saint Joseph University of Beirut, Beirut 111, Lebanon; 3CHU of Clermont-Ferrand, Service d’Odontologie, 63003 Clermont-Ferrand, France

**Keywords:** older people, dementia, nutrition, oral health, number of posterior dental functional units, mid-upper arm circumference

## Abstract

Oral health indicators are usually collected to alert for the presence of infectious diseases, but the impact of poor oral health on the nutritional status of older people with dementia is often neglected. This study aims to explore the relationship between the number of posterior dental functional units (PFUs) and the anthropometric measure of malnutrition, the mid-upper arm circumference (MUAC), in older people with dementia while controlling for other variables, and to analyze whether the number of PFUs is considered when adjusting the texture of the food provided at mealtimes. A total of 103 individuals who were 70 years or older with dementia were recruited from seven institutions. Cognitive impairment was assessed using the Mini Mental State Examination. Data were collected from medical records (sociodemographic characteristics, presence of chronic diseases, prescribed medications, results of serum albumin level), as well as questionnaires (type of feeding routes, type of food texture, supplements intake, Activity of Daily Living index), clinical examinations (MUAC), and oral health parameters (PFUs, oral dryness, oral hygiene). MUAC was the dependent outcome variable. MUAC was associated with the number of PFUs (*p =* 0.032); participants with PFU ≤ 4 were 7.5 times more likely to have MUAC < 21 cm than others. Other associations were found between MUAC and albumin level (OR = 12.5; *p =* 0.001), modified food texture (OR = 4.2; *p =* 0.035), and length of institutional stay (OR = 5.2; *p =* 0.033); however, the type of oral feeding was not significantly related to the number of PFUs (*p =* 0.487) so there is an inadequate correlation between food texture and oral health status. Similar to MUAC, the number of PFUs could be an oral anthropometric criterion that is recorded during routine hygiene care to alert for the risk of malnutrition and the inappropriate choice of food texture in older individuals with dementia.

## 1. Introduction

Since the number of older individuals is increasing, it has been estimated that the number of subjects with dementia will also increase worldwide—to 115.4 million by 2050 [1,2]. As dementia progresses, severe weight loss becomes a serious problem, especially among older subjects living in institutions [3]. Studies have shown that patients with cognitive problems experience greater weight loss compared with cognitively functioning older subjects, and are more at risk of malnutrition due to eating and swallowing difficulties [4,5]. The etiology of malnutrition in patients with dementia is multifactorial; potential contributing factors include the inability to prepare and eat food, [6] impairment of olfaction and taste, agitation, the presence of co-morbid medical illnesses [7], and oral health problems [8,9]. 

Oral health indicators are usually collected to alert healthcare workers of infectious diseases and pain during general health screening, but the impact of poor oral health on nutritional status is often neglected in older people with dementia. The challenges of caring for older people with dementia involve both health and social concerns. Self-perceived indicators cannot be used to provide relevant information about their oral health problems and the risk of malnutrition. Similarly, there is a tendency to avoid invasive examinations due to the difficulty of adapting these examinations to people with dementia, as changes in routine habits could induce anxiety and increase behavioral disorders. Given the risk of malnutrition and the need to limit medical visits and/or invasive examinations, there is a need to use valid indicators of malnutrition and oral health status that can be easily collected by nurses and caregivers. It has been previously reported that the mid-upper arm circumference (MUAC) is a good indicator of malnutrition for a wide range of health conditions. It has been used to identify nutritional risk in community-dwelling older adults [10,11,12,13,14] older people living at home [15,16], older people receiving in-home care [17] people attending health clinics [18] hospitalized patients with cancer [19,20] and older people with dementia [21,22]. These studies revealed that this indicator is a valid and easy-to-use tool in older subjects because it is simple, quick, non-invasive, and inexpensive.

However, oral status acts as a major factor in mastication performance and, in turn, has implications on the choice of food texture and feeding routes [23]. The use of anthropological criteria to identify both nutritional risk and oral condition could optimize the regular examinations of older subjects with dementia and could provide complementary information for adjusting the dietary texture based on their malnutrition risk and chewing ability. 

The number of posterior functional dental units (PFUs) is a non-invasive anthropometric criterion that is clearly related to chewing performance and has an impact on the nutritional status of older people [24,25,26,27,28]. This criterion needs to be considered in order to specify the most appropriate food texture and/or feeding route for each individual. Currently, it is not known whether the number of PFUs is related to the risk of malnutrition in older subjects with cognitive impairment, or if this indicator is considered when adjusting the texture of the food during meals. Therefore, this study was designed (1) to explore the relationship between the number of PFUs and the anthropometric measure of malnutrition, the mid-upper arm circumference (MUAC), in older people with dementia while controlling for other variables, and (2) to analyze whether the number of PFUs is taken into consideration when adjusting the texture of the food provided at mealtimes.

## 2. Materials and Methods

### 2.1. Design

This was a cross-sectional observational study. Ethical approval was obtained from the ethical research committee of Saint Joseph University of Beirut, Lebanon (Tfem/2018/68). Informed consent was obtained from participants and/or their legal representatives when residents were unable to respond due to their cognitive impairment. 

### 2.2. Sample

Participants were recruited from seven institutions in Beirut, Lebanon. During a six-month period (May 2018–15 November 2018), older patients with cognitive impairment who were 70 years or older and who had been living in the institution for at least three months were considered for inclusion in the study. 

The severity of cognitive impairment, based on the Mini Mental State Examination (MMSE), was used as an inclusion criterion. This examination includes tests of orientation in time and place, simple and complex attention, memory, language skills, and visual construction. It is a brief, structured 30-item questionnaire with scores ranging from 0 to 30. The classification of the severity of cognitive impairment was interpreted as follows: no cognitive impairment for scores between 24 and 30; mild cognitive impairment for scores between 19 and 23; moderate cognitive impairment for scores between 10 and 18; and severe cognitive impairment for scores below 10. Since scores above 24 are considered “normal”, only patients with scores below 24 were included in the study [29,30]. Residents with a terminal illness, as indicated by the staff at the institutions (*n =* 3), those with missing data (*n =* 8), and those for whom clinical examinations or anthropometric measurements were not possible (*n =* 4) were excluded from the study. 

### 2.3. Procedure

Data were collected from medical records, questionnaires, and oral and general health clinical examinations. 

### 2.4. Measures

The sociodemographic characteristics of the participants were collected from their medical records and included information on their age, gender, length of stay, and marital status. The medical records also provided information on the presence of chronic diseases and prescribed medications. The results of biological examinations performed in the previous few months to assess nutritional status were evaluated using data from the patients’ medical records. Older patients with a serum albumin concentration <35 g/L and with C-reactive protein less than 15 mg/dL were considered to have an inadequate nutritional status [31,32]. Additional information was collected from the institution’s dietician on the type of feeding route (oral or enteral), food texture (normal or modified texture: liquid, pureed, minced/moistened, or softened), and presence of dietary supplements. Nurses were asked about each participant’s level of autonomy. Clinical examinations for anthropometric measurements and oral health parameters were also performed.

*ADL index:* The Activity of Daily Living (ADL) index was used to assess the level of autonomy. This index includes questions on six self-care functions: bathing, dressing and toileting, transferring, continence, and feeding. The response for each function is scored as 0 for total dependence, 1 for total independence, and 0.5 for partial independence. The total ADL score ranges from 0 to 6, with 0 indicating that the participant is highly dependent and 6 being completely independent; thus, a score between 0 and 2 indicates “severe dependence”; a score between 2.5 and 4 indicates “moderate dependence”; and a score of 4.5 and above indicates “no to mild” dependence [33].

*Nutritional status:* The mid-upper arm circumference (MUAC) is an anthropometric indicator of nutritional status. It was measured on the upper left arm by a calibrated examiner at the midpoint between the tip of the shoulder and the tip of the elbow using a flexible non-elastic tape measure. Patients with a MUAC of less than 21 cm were considered to have an inadequate nutritional status [19,34]. 

*Oral health:* Dental screening was performed in the patients’ rooms at the nursing home by a qualified examiner. Basic dental instruments (mirror, gauze, and probe) were used in combination with a bright examination light. The number of PFUs, oral dryness, and oral hygiene status were evaluated.

Number of posterior functional units (PFUs): (a) For residents with natural teeth or fixed prostheses, and for those who had worn their dentures during the previous two meals, masticatory function was assessed using the number of PFUs by asking the participants to chew 1–2 cycles on 200 µm thick articulating paper. (b) For residents with dentures who were not using them during the last two meals, the number of PFUs was assessed without dentures. (3) For non-cooperating residents, the jaw was gently manipulated to the extent of inter-arch dental contacts. As a result, the number of molars and premolars (either natural or prosthetic) on the mandibular arch that had at least one colored mark gave the number of PFUs (Figure 1); however, the number of posterior teeth in the maxilla with colored marks is not counted as this examination is less easy to carry out. The number of PFUs ranges between 0 and 10, and masticatory efficiency is considered to be affected when it is less than or equal to 4 [23,26,27,35].

Oral dryness: The dryness of the mouth was evaluated using clinical signs such as depapillated and/or dry mucosa; mirror sticking to the buccal mucosa; scant, thick, and filamentous saliva; or dry floor of the mouth [36,37]. The presence of oral dryness was determined when at least one clinical sign of oral dryness was present. 

Dental/denture hygiene: (a) For dentate individuals, dental hygiene was determined using the modified Visible Plaque Index [38] and was evaluated on the upper and lower molar and premolar region and the upper and lower incisor region. For each region, dental hygiene was recorded for the crown and visible part of the root as being “good” for clean or almost clean teeth; “moderate” when dental plaque, calculus, or food remnants covered no more than one-third of the surface; and “poor” when dental plaque, calculus, or food remnants covered more than one-third of the surface. (b) Denture hygiene for denture wearers was assessed by examining the mucosal surface of the existing upper and/or lower denture. It was rated as “good” when the surface against the mucosa was clean or almost clean; “moderate” when dental plaque, calculus, or food remnants covered no more than one-third of the surface; and “poor” when dental plaque, calculus, or food remnants covered more than one-third of the surface [39]. (c) For edentulous patients without dentures, the oral hygiene criteria were the presence of food sticking on the palatal mucosa and/or on the vestibule. The subject’s overall oral hygiene was designated as “good”, “moderate”, or “poor” based on the maximal finding of denture, oral, and/or dental hygiene.

### 2.5. Data Analyses 

#### 2.5.1. Hypotheses

The first hypothesis was that malnutrition in older people with dementia might be related to their dental status. In particular, people presenting a low number of PFUs might be more likely to have a poor nutritional status than those with a good and functional dental status. Given the high prevalence of dementia in the target group, malnutrition and dental status were measured using anthropological indicators, MUAC, and the number of PFUs, respectively. This also involved corroborating whether the anthropometric measurement of malnutrition through MUAC is correlated with biological albumin levels. The second hypothesis was that the choice of the diet texture is influenced by the dental status of the residents. 

#### 2.5.2. Statistical Analyses

Statistical analyses were carried out using IBM SPSS Statistics version 27.0 software. The alpha error was set at 0.05. Sociodemographic characteristics, nutritional status, feeding route and texture, and general and oral health condition were described for the entire sample—and were adapted to the severity of dementia. The mean and standard deviations were calculated for continuous variables, and the frequency and percentage were calculated for the categorical variables. Kolmogorov–Smirnov tests were used to assess the normality distribution of continuous variables. The Chi-square test of independence or Fisher’s Exact test, and the Student’s t-test or its non-parametric equivalent, the Mann–Whitney test, were applied to compare categorical and continuous variables respectively according to the severity of dementia.

To evaluate the first hypothesis, statistical analyses were performed to assess the relationship between MUAC (as the outcome variable) and the number of PFUs (as the explanatory variable), while controlling for sociodemographic characteristics, feeding habits, general and oral health status, and albumin levels. The sample size (*n*) was calculated using the formula that considers the number of independent variables to be included in the model: *n =* 50 + 8 m, where m is the number of independent variables. Given that m is equal to 6 in the final model, a minimum of 98 subjects had to be included in the study [40]. Univariate analyses of categorical and continuous variables were carried out using the Chi-square test of independence or Fisher’s Exact test, and the Student’s t-test or its equivalent non-parametric test, the Mann–Whitney test, respectively. Logistic regression analysis was applied using the dichotomized MUAC values (<21 cm; ≥21 cm) as the dependent variable. Sociodemographic variables, feeding habits, general health parameters, and oral health variables that showed associations with a *p*-value < 0.200 in the univariate analyses were candidates for the multivariate model, according to the Enter method. In addition, albumin levels were also included in the model to attest its positive relationship with MUAC. Collinearity between independent variables was also tested using Spearman’s rank correlation coefficient, and highly correlated variables were excluded from the model—the type of oral feeding and ADL were highly correlated and were not included in the same model. 

To verify the second hypothesis, statistical analyses were conducted to assess the relationship between the type of diet texture and the number of PFUs and to identify the reasons for administering a “normal” or modified food texture to this population of older subjects with dementia. The Chi-square test of independence or Fisher’s Exact test and the Student’s t-test or its equivalent non-parametric test, the Mann–Whitney test, were applied to compare categorical and continuous variables, respectively.

## 3. Results

### 3.1. Characteristics of the Participants

One hundred and three participants (73.8% women) who were 83.90 ± 8.74 years old were included. The majority of participants were single, widowed, or divorced (82.2%). Seventy-two participants (69.9%) had lived in the institution for more than one year. The mean number of chronic diseases and medications consumed per day was 2.25 ± 1.53 and 7.54 ± 3.83, respectively. Sixty-eight (66.0%) participants had severe dementia; the mean MMSE score was 9.9 ± 3.1 and the mean ADL score was 1.53 ± 1.86. In terms of feeding behavior, 29.2% consumed texture-modified foods and 9.7% took dietary supplements. In terms of their nutritional status, 24.3% had a MUAC of less than 21 cm, and 34% had hypoalbuminemia. Regarding their oral health status, 76.7% had clinical signs of dry mouth and 64.1% had poor oral hygiene or had four PFUs or fewer. The distributions of the descriptive variables according to the severity of dementia are presented in Table 1. Overall, more patients with severe dementia were less autonomous, had hypoalbuminemia, and ate food with a modified texture.

### 3.2. Anthropometric Measurements of Malnutrition in the Study Group

Albumin levels were significantly associated with MUAC (*p* < 0.001) (Table 2 and Table 3); participants with albumin levels < 35 g/L were 12.5 times more likely to have MUAC < 21 cm than others (Table 3). This demonstrates that measuring malnutrition with MUAC is reliable for this study group.

### 3.3. Variables Associated with Malnutrition as Measured by MUAC

*Oral health condition:* Participants with PFU ≤ 4 were 7.5 times more likely to have MUAC < 21 cm than those with PFU > 4 (*p =* 0.032) (Table 3). Dry mouth was significantly related to MUAC in the univariate analysis (*p =* 0.038) (Table 2), although this was not the case for the multivariate analysis (*p =* 0.159) (Table 3).

*Sociodemographic characteristics:* Age (*p =* 0.682), gender (*p =* 0.417), and marital status (*p =* 0.551) were not significantly related to MUAC (Table 2); however, multivariate analysis revealed that participants with a length of stay of one year or less were 5.2 times more likely to have a lower MUAC than participants who had lived in the institution for more than one year (*p =* 0.029) (Table 3).

*General health condition:* The number of medications taken (*p =* 0.384) and the number of chronic diseases (*p =* 0.346) were not significantly associated with MUAC; however, the mean ADL score was significantly higher in participants with MUAC ≥ 21 cm compared to those with MUAC < 21 cm (Table 2). 

*Feeding habits:* The type of oral feeding was significantly associated with MUAC (*p*-value = 0.035) (Table 3); participants with a texture-modified diet were 4.2 times more likely to have MUAC < 21 cm than those with a normal-textured diet, whereas the feeding route (*p =* 0.256) and dietary supplements (*p =* 0.251) were not significantly associated (Table 2).

### 3.4. Variables Associated with Type of Oral Feeding 

The type of oral feeding was significantly associated with nutritional status (*p* = 0.001), dietary supplement intake (*p =* 0.017), severity of dementia (*p =* 0.008), and ADL score (*p* < 0.001). A higher percentage of subjects on a texture-modified diet had MUAC < 21 cm, albumin levels < 35 g/L, were taking dietary supplements, and had severe dementia compared with the corresponding percentage of subjects on a normal-texture diet. The mean ADL score was significantly lower in participants on texture-modified diet; however, the type of oral feeding was not significantly related to the number of PFUs (*p =* 0.487), dry mouth (*p =* 0.300), or oral hygiene (*p =* 0.817) (Table 4). 

Furthermore, the results revealed that 57 participants with dry mouth had PFU ≤ 4. Among them, 34 (59.6%) were on a normal-texture diet, 17 (29.8%) were on a modified-texture diet, and 6 (10.5%) had an enteral feeding route. This also confirms that there is an inadequacy between food texture and oral health status.

## 4. Discussion

This study reports that the number of PFUs was linked to MUAC in a group of older subjects with dementia, for whom MUAC and albumin levels were highly correlated. Participants with four or fewer PFUs were 7.5 times more likely to suffer from malnutrition, compared to those with more than four PFUs. These results confirm our first hypothesis, which stated that the number of PFUs could be used in association with MUAC to alert for nutritional deprivation and chewing difficulties in older subjects with cognitive deficiencies or dementia who are unable to communicate their needs. The clinical measurement of the number of PFUs is a reliable indicator of masticatory function, which determines masticatory capacity [23,26,35].

However, the second hypothesis was not confirmed, as the number of PFUs was not statistically associated with the texture of the administered food. Indeed, 66% of the individuals had an oral diet of normal consistency, although the mean number of PFUs was 3.5 ± 3.3. In addition, 60% of the participants with a normal diet had PFU ≤ 4 and were therefore unable to chew effectively. Hence, these results exhibit that the type of diet is not adjusted according to the chewing capacities of the participants. Thus, oral examinations during regular check-ups in older subjects are essential to ensure an appropriate choice of food texture. 

During clinical examinations, measuring MUAC in association with the number of PFUs provided information on nutritional status, in addition to the essential need to adjust food texture to the oral functional status of the resident. To ensure safe oral feeding, the decision-making process for individuals with poor oral health must consider either dental treatment or texture modification of their diet. For those people who can afford it, it may be more appropriate to seek oral rehabilitation treatment to maintain mastication and swallowing function [41,42]; however, this presents a challenge for older adults with cognitive impairment. Three factors have a major impact on masticatory function in older persons: impaired motor apparatus function, quantity and/or quality of saliva, and number of natural antagonist teeth [23,35]. In most cases, older subjects with cognitive impairments have problems with all three factors. In this study, 42 (41%) individuals had a high degree of motor problems with an ADL score of ≤2, in addition to chewing incapacities (PFU ≤ 4) and oral dryness. Of these 42 individuals, 20 (47%) had a “normal” diet with no blended food. In contrast, one person with eight PFUs who had no signs of a dry mouth and a mild motor impairment (ADL score of four or more) was on a blended food diet. Clearly, the decision to administer blended food must be supported by reliable criteria, such as MUAC and PFU values.

Based on albumin levels, 34.0% of the participants were malnourished and had an albumin level <35 g/L, of which 6.8% were severely malnourished—with an albumin < 30 g/L. In addition, 24.3% had a MUAC < 21 cm. These results show that the prevalence of malnutrition was high in this study group compared to older people living in institutions with no cognitive deficiency (3.2%) [29], or those with mild to moderate dementia (12.6%) [43]. It is therefore important to perform nutritional screening to assess patients with dementia at the time of diagnosis, and to monitor changes in MUAC and PFU over time, especially when the majority of patients that are newly diagnosed with dementia may have a normal nutritional status and better oral health [44].

Our study indicates that oral health status is neglected in older people with dementia, which is consistent with other studies [45,46,47,48]. A total of 64.1% had poor oral hygiene, 76.7% had signs of dry mouth, and 64.1% had a low number of PFUs. When older adults become care-dependent, their oral health usually worsens and receives less attention. In fact, mouth dryness has often been reported as a common adverse effect of cerebral tropic drugs [49]. Similarly, oral health conditions, such as periodontitis and dryness of the mouth are associated with an increased risk of cardiovascular disease and type 2 diabetes mellitus [50]. Caregivers for older residents are reluctant to perform oral health care due to their limited knowledge [51,52,53]. Sometimes oral health indicators are collected to alert for infectious diseases or pain, but the impact of poor oral health on nutritional status is usually overlooked. Increasing the care staff’s knowledge about the impact of oral status on nutrition could positively change their view on daily oral hygiene. The number of PFUs could easily be measured by dental nurses and dental therapists, as well as by physicians, nurses, families, and caregivers. Caregivers and their families may need guidance, firstly on oral health and nutritional status and, secondly, on how to measure the number of PFUs. For non-dental professionals who are not confident about carrying out this procedure, a less sensitive count of the number of posterior biting teeth could be performed during the daily oral hygiene routine. It should be noted that according to some establishments’ protocols, in order to simplify the work of the caregivers, denture wearers are systematically deprived of their dentures, regardless of the efficiency and quality of the dentures. Obviously, there is a real need to increase awareness of oral health and nutrition among the general population.

### Limitations

This is the first study to assess the association between oral health parameters and nutritional status in older people with severe cognitive impairment; however, some limitations need to be addressed. Firstly, this exploratory cross-sectional study cannot determine a causal relationship between anthropometric oral health indicators and malnutrition. Further longitudinal studies are needed to assess the temporal relationship and to establish causality. With this in mind, a calibration process for non-dental professionals and caregivers is needed in order to count the number of PFUs. Secondly, the study was performed on a convenient sample of older people with dementia that were recruited from seven long-term care institutions. This study group does not represent the population with dementia, and therefore our findings cannot be fully generalized to this population. Thirdly, a larger sample size is needed in order to draw valid conclusions and to obtain results that are more precise. Fourthly, the food types (protein, fat, carbohydrate) of the patients’ diets were not assessed. It has been reported that subjects over 65 years old have an insufficient energy intake as well as deficits in vitamins and micronutrients, regardless of their oral status (dentate or denture wearers) [54]. This might interfere with and confound the link between MUAC and the other independent variables.

## 5. Conclusions

Older people living in institutions and suffering from cognitive decline represent a fragile population with a high risk of malnutrition, and whose meals are often of inappropriate texture. 

Routine screening for dementia and regular nutritional assessment should be an integral part of care in long-term care institutions. Our results, despite their limitations, revealed that the number of PFUs could be an anthropometric criterion, as could MUAC, and that this information could be collected during routine care procedures to alert to inconsistent food textures and the risk of malnutrition in elderly subjects with dementia. Further prospective studies should be performed on a larger sample to confirm these findings.

## Figures and Tables

**Figure 1 ijerph-19-15086-f001:**
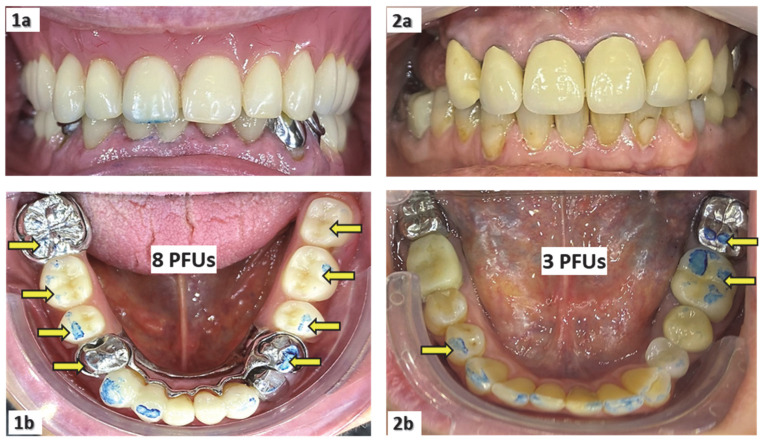
Measurements of the number of posterior functional dental units (PFUs) for a patient using his upper full and lower partial dentures during the previous two meals (**1a**,**1b**), and for another patient who did not use his upper partial denture during the previous two meals (**2a**,**2b**).

**Table 1 ijerph-19-15086-t001:** Description of the sociodemographic characteristics, nutritional status, feeding behavior, and general and oral health status according to the severity of dementia.

		Severity of Dementia			
		Moderate (*n =* 35)	Severe (*n =* 68)	Total	*p*-Value
**Sociodemographic characteristics**	**Age** (years)	84.69 ± 8.64	83.49 ± 8.83	83.90 ± 8.74	0.515
	**Gender**				0.388
	*Men*	11 (31.4%)	16 (23.5%)	27 (26.2%)	
	*Women*	24 (68.6%)	52 (76.5%)	76 (73.8%)	
	**Marital status**				0.201
	*Single*	6 (17.1%)	15 (22.1%)	21 (20.4%)	
	*Married*	5 (14.3%)	13 (19.1%)	18 (17.5%)	
	*Divorced/widowed*	24 (68.6%)	40 (58.8%)	64 (62.2%)	
	**Duration of stay in institution**				0.487
	*<*1 *year*	9 (25.7%)	22 (32.4%)	31 (30.1%)	
	*>*1 *year*	26 (74.3%)	46 (67.6%)	72 (69.9%)	
**Nutritional status**	**MUAC < 21 cm**	5 (14.3%)	20 (29.4%)	25 (24.3%)	0.090
	**Albumin < 35 g/L**	7 (20.0%)	28 (41.2%)	35 (34.0%)	0.032
**Feeding routes and food texture**	**Feeding route**				0.092
	*Oral*	35 (100.0%)	61 (89.7%)	96 (93.2%)	
	*Enteral*	0 (0.0%)	7 (10.3%)	7 (6.8%)	
	**Food texture ***				0.015
	*Normal texture*	30 (85.7%)	38 (62.3%)	68 (70.8%)	
	*Modified texture*	5 (14.3%)	23 (37.7%)	28 (29.2%)	
	**Dietary supplement intake**	2 (5.7%)	8 (11.8%)	10 (9.7%)	0.488
**General health status**	**Drug intake number per day**	7.34 ± 3.82	7.65 ± 3.86	7.54 ± 3.83	0.705
	**Number of chronic diseases**	2.51 ± 1.31	2.12 ± 1.63	2.25 ± 1.53	0.215
	**ADL score**	2.39 ± 2.12	1.1 ± 1.56	1.53 ± 1.86	<0.001
**Oral health status**	**FU *≤* 4**	23 (65.7%)	43 (63.2%)	66 (64.1%)	0.804
	**Dry mouth**	24 (68.6%)	55 (80.9%)	79 (76.7%)	0.162
	**Poor oral hygiene**	19 (54.3%)	47 (69.1%)	66 (64.1%)	0.137

Mean ± Std Dev are used for continuous variables. Frequency (N) and percentage (%) are used for categorical variables. * Excluding subjects with enteral feeding route.

**Table 2 ijerph-19-15086-t002:** Univariate analyses of independent variables associated with MUAC.

		MUAC <21 cm (*n =* 25)	MUAC ≥21 cm (*n =* 78)	*p*-Value
**Sociodemographic characteristics**	**Age** (years)	84.40 ± 9.17	83.56 ± 8.74	0.682
	**Gender**			
	*Men*	5 (18.5%)	22 (81.5%)	0.417
	*Women*	20 (26.3%)	56 (73.7%)	
	**Marital status**			
	*Married*	3 (16.7%)	15 (83.3%)	0.551
	*Others*	22 (25.9%)	63 (74.1%)	
	**Duration of the stay**			
	*<*1 *year*	11 (35.5%)	20 (64.5%)	0.082
	*>*1 *year*	14 (19.4%)	58 (80.6%)	
**Feeding habits**	**Feeding route**			
	*Oral*	22 (22.9%)	74 (77.1%)	0.256
	*Enteral*	3 (42.9%)	4 (57.1%)	
	**Food texture** *			
	*Normal texture*	9 (13.2%)	59 (86.8%)	0.001
	*Modified texture*	13 (46.4%)	15 (53.6%)	
	**Dietary supplement intake**			
	*Yes*	4 (40.0%)	6 (60.0%)	0.251
	*No*	21 (22.6%)	72 (77.4%)	
**Nutritional status**	**Albumin level**			
	*<*35 g/L	19 (76.0%)	16 (20.5%)	<0.001
	*>*35 g/L	6 (24.0%)	62 (79.5%)	
**Oral health status**	**PFUs**			
	*≤*4	21 (31.8%)	45 (68.2%)	0.017
	*>*4	4 (10.8%)	33 (89.2%)	
	**Dry mouth**			
	*No*	2 (8.3%)	22 (91.7%)	0.038
	*Yes*	23 (29.1%)	56 (70.9%)	
**General health status**	**Drugs intake number**	6.96 ± 3.43	7.73 ± 3.95	0.384
	**Number of chronic diseases**	2.00 ± 1.29	2.33 ± 1.60	0.346
	**ADL score**	0.46 ± 0.88	1.88 ± 1.97	0.001

Mean ± Std Dev are used for continuous variables. Frequency (N) and percentage (%) are used for categorical variables. * Excluding subjects with enteral feeding route.

**Table 3 ijerph-19-15086-t003:** Logistic regression analysis of the association between MUAC as outcome variable and PFUs as explanatory variable, while controlling other variables.

	B	S.E.	Wald	Sig.	OR	95% C.I. for OR	
						Lower	Upper
Albumin level < 35 g/L	2.526	0.684	13.625	0.000	12.503	3.270	47.811
Duration of stay ≤ 1 year	1.643	0.750	4.792	0.029	5.169	1.188	22.495
Modified food texture	1.446	0.687	4.434	0.035	4.247	1.105	16.316
PFUs ≤ 4	2.015	0.941	4.589	0.032	7.504	1.187	47.438
Clinical sign of dry mouth	1.352	0.960	1.986	0.159	3.867	0.589	25.368
Constant	−10.507	2.812	13.960	0.000	0.000		

The following independent variables: ‘food texture’ and ‘ADL’ were highly correlated and were not included in this model.

**Table 4 ijerph-19-15086-t004:** Analysis of type of oral feeding with sociodemographic characteristics, nutritional status, feeding habits, and general health status and oral health status.

		Type of Oral Feeding		
		Normal Texture (*n =* 68)	Modified Texture (*n =* 28)	*p*-Value
**Sociodemographic characteristics**	Men	20 (29.4%)	5 (17.9%)	0.241
	Age	83.75 ± 8.77	83.32 ± 9.33	0.831
	Duration of stay > 1 year	49 (72.1%)	21 (75.0%)	0.768
**Nutritional status**	MUAC < 21 cm	9 (13.2%)	13 (46.4%)	0.001
	Albumin < 35 g/L	14 (20.6%)	15 (53.6%)	0.001
**Feeding habits**	Dietary supplement intake	3 (4.4%)	6 (21.4%)	0.017
**General health status**	Number of chronic diseases	2.38 ± 1.56	2.07 ± 1.36	0.359
	Drug intake number/day	7.82 ± 3.87	7.57 ± 3.12	0.760
	ADL	2.17 ± 1.94	0.34 ± 0.87	<0.001
	Severe dementia	36 (52.9%)	23 (82.1%)	0.008
**Oral health status**	Number PFUs ≤ 4	41 (60.3%)	19 (67.9%)	0.487
	Dry mouth	49 (72.1%)	23 (82.1%)	0.300
	Bad oral hygiene	43 (63.2%)	17 (60.7%)	0.817

Mean ± Std Dev are used for continuous variables. Frequency (N) and percentage (%) are used for categorical variables.

## Data Availability

Not applicable.

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
