# Peer review of "Dental Criteria Could Alert for Malnutrition Risk and Inappropriate Choice of Food Texture in Older Subjects with Dementia: An Analytical Cross-Sectional Study"

_ijerph, 2022, doi:10.3390/ijerph192215086_

Round 1
Reviewer 1 Report
Dear Authors,
your paper is well conducted and is very important to reenforce the concept of the importance of oral health, their correlations with the general health and the role of the dental hygienist in the hospitals and institutions.
I accept the paper in the present form.
Author Response
Thank you
Reviewer 2 Report
Dear authors,
I found the manuscript's theme interesting, actual and more than that neglected. The problem of improving the living conditions of the elderly population, especially of institutionalized individuals, that geriatric specialists face, is a subject that needs to be given more attention.
From social considerations, being considered an inactive population group, but especially economically - by not actively participating in state budgets, the elderly are somewhat ignored even in underdeveloped, moderately developed countries or in developing countries.
Thus, the health problems of the elderly are viewed from more palliative than curative perspectives in the sense of improving their living conditions.
In the Introduction Section, the subject is presented clearly, in a concise manner, touching on several real aspects faced by this population group, whether we refer to people cared for at home, independent or not, or to those who are institutionalized, being almost like an alarm signal for this population group, especially those institutionalized.
The impossibility of implementing a healthy diet, for reasons of cognitive deficit or due to restrictions caused by age-specific systemic conditions and especially oral health problems, can be interpreted by following some physical indicators associated with systemic blood markers, in this study being brought into discussion: MUAC, PFU and Albumin marker.
The manuscript is written in an organized manner, and is reader-friendly. In consequence, I have some minor suggestions:
● I suggest the reorganization of the Results section -in my opinion the first paragraph should be included in the Materials and Methods sections
● Being a cross section study, a conclusive analysis was not carried out on the association through conclusive data of the correlation between the systemic markers and those of the oral analysis, something that I consider necessary.
● I would also consider it necessary to add the problematic medication administered to this group, whether it is cerebral trophics, for cardiac problems or nutritional diseases in the analysis of symptoms.
Do you think that it could influence the oral manifestations in a negative way?
● You mentioned symptoms of dry mouth in the text. They can be caused either by diabetes or by specific medication for cognitive disorders.
If you haven't taken this aspect into account, important by the way, it can be the subject of a new investigation.
● It is surprising to me that these analyzed patients, mentioned with problems caused by nutrition, present natural odonto-periodontal units in the posterior areas and not only fixed or dentures.
● I return to the question asked previously on page 4 first paragraph Oral dryness: The dryness of the mouth was evaluated using clinical signs such as
depapillated and/or dry mucosa; mirror sticking to the buccal mucosa; scant, thick, and filamentous saliva; or dry floor of the mouth.[35,36] The presence of oral dryness was determined when at least one clinical sign of oral dryness was present.. It is possible that this oral condition is the cause of the medication and not of malnutrition
Result section and statistic analysis
Results are presented in a well structured text with accompanying tables. Statistical analysis covers multiple variables and correlates MUAC depending on key factors.
A more clear emphasis on results and statistical analysis outcome would be beneficial in understanding the correlations involved. More table explanatory text would be helpful.
Conclusion
Though it has some limitations that make it hard to generalize the results, this article highlights some criteria to assess malnutrition in institutionalized dementia patients. Further research on this topic could help improve the quality of life of the elderly suffering from dementia.
Please accept my sincere apologies for the delayed answer.
Author Response
ANSWERS TO REVIEWER 2
1- I suggest the reorganization of the Results section -in my opinion the first paragraph should be included in the Materials and Methods sections
The first paragraph of the results section describes the characteristics of the participants that might be reported in the Results section since it was displayed after the analyses of the data of the participants.
2- Being a cross section study, a conclusive analysis was not carried out on the association through conclusive data of the correlation between the systemic markers and those of the oral analysis, something that I consider necessary.
We agree with reviewer#1, as this is a cross-sectional designed study, it would not be appropriate to give conclusive results on the correlation between systemic markers and oral parameters. A longitudinal study is needed to assess the relationships and to establish causality between these markers. However, during the analysis of the results, the sample size was enough to analyze the relationship between systemic and oral markers. Considering the frailty of the subjects, and the consequent difficulties to repeat a longitudinal study we choose to optimize our cross-sectional study in analyzing the correlation. We discuss this point in the limitation section of the discussion.
This is the first study to assess the association between oral health parameters and nutritional status in older people with severe cognitive impairment. However, some limitations need to be addressed. Firstly, this exploratory cross-sectional study cannot determine a causal relationship between anthropometric oral health indicators and malnutrition. Further longitudinal studies are needed to assess the temporal relationship and to establish causality.
3- I would also consider it necessary to add the problematic medication administered to this group, whether it is cerebral trophics, for cardiac problems or nutritional diseases in the analysis of symptoms. Do you think that it could influence the oral manifestations in a negative way?
We agree with the reviewer comment. Xerostomia has often been reported as a common adverse effect of cerebral tropic drugs (Cockburn N, Pradhan A, Taing MW, Kisely S, Ford PJ. Oral health impacts of medications used to treat mental illness. J Affect Disord. 2017;223:184-193. doi:10.1016/j.jad.2017.07.037). In our study, we have collected information about the number of daily taken drugs, but not for the pharmacological categories of the medications. We found that the number of drugs did not impact oral status, this finding was also reported in a previous study (Paredes-Rodríguez VM, Torrijos-Gómez G, González-Serrano J, López-Pintor-Muñoz RM, López-Bermejo MÁ, Hernández-Vallejo G. Quality of life and oral health in elderly. J Clin Exp Dent. 2016;8(5):e590-e596. Published 2016 Dec 1. doi:10.4317/jced.53317). Again, a further longitudinal study would be necessary to analyze this relationship more finely.
We added the problematic medication administered to this group, in the Discussion section as follows:
Our study indicates that oral health status is neglected in older people with dementia, which is consistent with others studies.[45–48] 64.1% had poor oral hygiene, 76.7% had signs of dry mouth, and 64.1% had a low number of PFUs. When older adults become care-dependent, their oral health usually worsens and receives less attention. In fact, mouth dryness has often been reported as a common adverse effect of cerebral tropic drugs [49]. Similarly, oral health conditions, such as periodontitis and dryness of the mouth are associated with an increased risk of cardiovascular disease and type 2 diabetes mellitus [50].
4- You mentioned symptoms of dry mouth in the text. They can be caused either by diabetes or by specific medication for cognitive disorders. If you haven't taken this aspect into account, important by the way, it can be the subject of a new investigation.
We agree. The presence or the absence of clinical sign of oral dryness (whatever its reason: age, medication, chronic diseases) was the controlled variables of our study to assess the relationship between PFUs and MUAC. The reason of the oral dryness was not evaluated. This information was already added in the discussion section as in point 3.
Our study indicates that oral health status is neglected in older people with dementia, which is consistent with others studies.[45–48] 64.1% had poor oral hygiene, 76.7% had signs of dry mouth, and 64.1% had a low number of PFUs. When older adults become care-dependent, their oral health usually worsens and receives less attention. In fact, mouth dryness has often been reported as a common adverse effect of cerebral tropic drugs [49]. Similarly, oral health conditions, such as periodontitis and dryness of the mouth are associated with an increased risk of cardiovascular disease and type 2 diabetes mellitus [50].
5- It is surprising to me that these analyzed patients, mentioned with problems caused by nutrition, present natural odonto-periodontal units in the posterior areas and not only fixed or dentures.
We described the oral status of the sample. Indeed, depending on the countries, and the social context of older persons some could have preserved natural teeth.
6- I return to the question asked previously on page 4 first paragraph Oral dryness: The dryness of the mouth was evaluated using clinical signs such as depapillated and/or dry mucosa; mirror sticking to the buccal mucosa; scant, thick, and filamentous saliva; or dry floor of the mouth.[35,36] The presence of oral dryness was determined when at least one clinical sign of oral dryness was present.. It is possible that this oral condition is the cause of the medication and not of malnutrition
We agree. In our study we did not analyze the reason of oral condition and that medication causes poor oral status. The oral condition described by the number of PFUs as the explanatory variable, the oral dryness as the controlled variable and the MUAC as the outcome variable of the study. This information was already added in the discussion section as in point 3.
Our study indicates that oral health status is neglected in older people with dementia, which is consistent with others studies.[45–48] 64.1% had poor oral hygiene, 76.7% had signs of dry mouth, and 64.1% had a low number of PFUs. When older adults become care-dependent, their oral health usually worsens and receives less attention. In fact, mouth dryness has often been reported as a common adverse effect of cerebral tropic drugs [49]. Similarly, oral health conditions, such as periodontitis and dryness of the mouth are associated with an increased risk of cardiovascular disease and type 2 diabetes mellitus [50].
7- Results are presented in a well-structured text with accompanying tables. Statistical analysis covers multiple variables and correlates MUAC depending on key factors.
A more clear emphasis on results and statistical analysis outcome would be beneficial in understanding the correlations involved. More table explanatory text would be helpful.
For table 1, an explanatory sentence was added as follow:
Participants with albumin level<35g/l were 12.5 times more likely to have MUAC<21cm than others (Table 3). Overall, more patients with severe dementia are less autonomous, had hypoalbuminemia and eat food with a modified texture.
Tables 2 and 3 need to be interpreted concomitantly. We added the following sentence:
Anthropometric measurements of malnutrition in the study group
Albumin levels were significantly associated with MUAC (p<0.001) (Tables 2 and 3). Participants with albumin level<35g/l were 12.5 times more likely to have MUAC<21cm than others (Table 3). This demonstrates that measuring malnutrition with MUAC is reliable for this study group.
The place of tables 2 and 3 were edited and adjusted in line to its corresponding text. This will improve their interpretation with the present text.

Reviewer 3 Report
Good job!
Moderate english corrections required.
Author Response
Moderate English corrections required.
English language was revised as requested.
Reviewer 4 Report
This study aims to assess the association of dental status (posterior functional units) with the nutritional status (mis-upper-arm circumference) among the institutionalised older people suffering from dementia. The authors have also tried to evaluate whether the dental status is considered before appropriate food texture is recommended to this target group. The authors have focussed on a vulnerable target group and have presented some very important findings in this novel research. Yet, the following corrections are suggested to present this study in a more comprehensive manner:
1. Title:
The title of the study has stated two terminologies that needs to be re-considered: the term ‘inconsistent’ is not specified anywhere in the article, nor is this the objective of the study. Hence, it is creating confusion. Rather, the secondary objective of this study is to evaluate whether dental status is considered before specifying the type of food texture that should be consumed. The authors should replace the term ‘inconsistent’ with a better suited term.
Secondly, the study design stated is cross-sectional observational study. The authors have assessed the association of dental status with malnutrition., which makes it a cross-sectional analytical study.
2. Abstract:
The abstract should start with a brief background (as is also mentioned in the instructions to the authors of this journal).
In the methods, all the independent and dependent variables assessed should be mentioned.
The stated conclusion is more like the study recommendation. It is suggested to reword the conclusion in line with the study objectives.
3. Methods:
The authors have gathered data on most of the factors that might have an influence on malnutrition. However, there is no mention about the dietary chart of these patients. The study focusses only on the food texture, but the food type (protein, fat, carbohydrate) might also have a confounding role on the dependent variable (mid upper arm circumference). The authors may comment on the food type in the discussion/limitation.
4. Results:
Page 8, Feeding habits: The results show that participants with a texture-modified diet were 4.2 times more likely to have MUAC<21cm than those with a normal textured diet. This shows that people who are on the modified-textured diet (a diet that needs to be recommended to older subjects having PFU<4) are more vulnerable to malnutrition than those individuals consuming normal diet. This is in total contrast to the hypothesis and the recommendation stated in this study. The results drawn for feeding habits may be due to the disproportionate number of subjects in the normal-texture and modified textured categories. Moreover, the sample size of this study is already very small. The authors should comment on this in the discussion section so that the hypothesis, the results and the recommendation are all in line.
5. Discussion:
Page 9, 2nd paragraph: The authors have discussed the results of the secondary objective, i.e., association of number of PFUs with the texture of administered food. However, these results have not been mentioned in the text/ table in the Results section. Since this is one of the objectives, it is essential to state the results, even if they are not significant.
Author Response
ANSWERS TO REVIEWER 4
- Title: The title of the study has stated two terminologies that needs to be re-considered: the term ‘inconsistent’ is not specified anywhere in the article, nor is this the objective of the study. Hence, it is creating confusion. Rather, the secondary objective of this study is to evaluate whether dental status is considered before specifying the type of food texture that should be consumed. The authors should replace the term ‘inconsistent’ with a better suited term. Secondly, the study design stated is cross-sectional observational study. The authors have assessed the association of dental status with malnutrition., which makes it a cross-sectional analytical study.
The title was adjusted according to the reviewer comment.
Dental criteria could alert for malnutrition risks and inappropriate choice of food texture in older subjects with dementia: An analytical cross-sectional study
- Abstract: The abstract should start with a brief background (as is also mentioned in the instructions to the authors of this journal).
In the methods, all the independent and dependent variables assessed should be mentioned. The stated conclusion is more like the study recommendation. It is suggested to reword the conclusion in line with the study objectives.
The abstract was adjusted according to the reviewer comments: we have added a brief background. All the dependent and independent variables were included in the methods section. Conclusion was reworded in line with the study objectives.
ABSTRACT: Background: Oral health indicators are usually collected to alert for the presence of infectious diseases, but the impact of poor oral health on the nutritional status of older people with dementia is often neglected. Aim: To explore the relationship between the number of Posterior dental Functional Units (PFUs) and the anthropometric measure of malnutrition, the Mid Upper Arm Circumference (MUAC), in older people with dementia while controlling for other variables, and to analyze whether the number of PFUs is considered when adjusting the texture of the food provided at mealtimes. Methods: 103 individuals aged 70 years or more with dementia were recruited from seven institutions. Cognitive impairment was assessed using the Mini Mental State Examination. Data were collected from medical records (sociodemographic characteristics, presence of chronic diseases, prescribed medications, results of serum albumin level), as well as questionnaires (type of feeding routes, type of food texture, supplements intake, Activity of Daily Living index), clinical examinations (MUAC) and oral health parameters (PFUs, oral dryness, oral hygiene). The MUAC was the dependent outcome variable. Results: MUAC was associated with the number of PFUs (p=0.032); participants with PFU ≤4 were 7.5 times more likely to have MUAC <21cm than others. Other associations were found between MUAC and albumin level (OR=12.5; p=0.001), modified food texture (OR=4.2; p=0.035) and length of institutional stay (OR=5.2; p=0.033). However, the type of oral feeding was not significantly related to the number of PFUs (p=0.487), so there is an inadequacy between food texture and oral health status. Conclusion: Similar to MUAC, the number of PFUs could be an oral anthropometric criterion recorded during routine hygiene care to alert for the risk of malnutrition and the inappropriate choice of food texture in older individuals with dementia.
- Methods: The authors have gathered data on most of the factors that might have an influence on malnutrition. However, there is no mention about the dietary chart of these patients. The study focusses only on the food texture, but the food type (protein, fat, carbohydrate) might also have a confounding role on the dependent variable (mid upper arm circumference). The authors may comment on the food type in the discussion/limitation.
The reviewer presented a relevant information. In this study, the dietary daily chart of patients was not assessed. In the limitation section, we have added the following sentence.
Fourthly, the food types (protein, fat, carbohydrate) of the patients’ diet were not assessed. It has been reported that subjects over 65 years old have an insufficient energy intake as well as deficits in vitamins and micronutrients, regardless of their oral status (dentate or denture wearers) [54]. This might interfere with and confound the link between MUAC and the other explanatory variables.
- Results: Page 8, Feeding habits: The results show that participants with a texture-modified diet were 4.2 times more likely to have MUAC<21cm than those with a normal textured diet. This shows that people who are on the modified-textured diet (a diet that needs to be recommended to older subjects having PFU<4) are more vulnerable to malnutrition than those individuals consuming normal diet. This is in total contrast to the hypothesis and the recommendation stated in this study. The results drawn for feeding habits may be due to the disproportionate number of subjects in the normal-texture and modified textured categories. Moreover, the sample size of this study is already very small. The authors should comment on this in the discussion section so that the hypothesis, the results and the recommendation are all in line.
We agree with reviewer#4, as written in the discussion section, the results confirmed our first hypothesis, but not the second one. The participants with a texture-modified diet were 4.2 times more likely to have MUAC<21cm than those with a normal textured diet. The results drawn for feeding habits may be due to the fact that a higher percentage of subjects on a texture-modified diet had severe dementia and have a lower ADL score compared with the corresponding percentage of subjects on a normal texture diet (as reported in Table 4). This elucidate that people who are on the modified-textured diet were more vulnerable to malnutrition because the majority had severe dementia and have a lower ADL score than those individuals consuming normal diet.
Our first hypothesis which states that people presenting a low number of PFUs might be more likely to have poor nutritional status than those with good functional dental status was confirmed when adjusting for other controlled variables; the type of the diet texture was the controlled variable in this relationship.
Our second hypothesis states about positive relationship between the type of diet and the number of posterior functional pairs of teeth, to identify the reasons for administering a “normal” or modified diet texture to this population of older subjects with dementia. However, the results showed that the type of oral feeding was not significantly related to the number of PFUs (p=0.487) and dry mouth (p=0.300). So, the choice of the food texture was never considered. Therefore, the second hypothesis was not confirmed, namely that the number of PFUs was not associated with the diet texture.
- Discussion: Page 9, 2ndparagraph: The authors have discussed the results of the secondary objective, i.e., association of number of PFUs with the texture of administered food. However, these results have not been mentioned in the text/ table in the Results section. Since this is one of the objectives, it is essential to state the results, even if they are not significant.
The results of the second hypothesis concerning the association between the number of PFUs and the texture of administered food were presented in the Results section as well as on Table 4.
Variables associated with type of oral feeding
However, the type of oral feeding was not significantly related to the number of PFUs (p=0.487), dry mouth (p=0.300) or oral hygiene (p=0.817) (Table 4).
Table 4. Analysis of type of oral feeding with sociodemographic characteristics, nutritional status, feeding habits, general health status and oral health status.
|
|
Type of oral feeding |
|||
|
Normal texture (n=68) |
Modified texture (n=28) |
-p-value |
||
|
Sociodemographic characteristics |
Men |
20(29.4%) |
5(17.9%) |
0.241 |
|
Age |
83.75±8.77 |
83.32±9.33 |
0.831 |
|
|
Duration of stay > 1 year |
49(72.1%) |
21(75.0%) |
0.768 |
|
|
Nutritional status |
MUAC < 21cm |
9(13.2%) |
13(46.4%) |
0.001 |
|
Albumin < 35g/l |
14(20.6%) |
15(53.6%) |
0.001 |
|
|
Feeding habits |
Nutritional supplements |
3(4.4%) |
6(21.4%) |
0.017 |
|
General health status |
Number of chronic diseases |
2.38±1.56 |
2.07±1.36 |
0.359 |
|
Drug intake number/day |
7.82±3.87 |
7.57±3.12 |
0.760 |
|
|
ADL |
2.17±1.94 |
0.34±0.87 |
<0.001 |
|
|
Severe dementia |
36(52.9%) |
23(82.1%) |
0.008 |
|
|
Oral health status |
Number PFUs ≤ 4 |
41(60.3%) |
19(67.9%) |
0.487 |
|
Dry mouth |
49(72.1%) |
23(82.1%) |
0.300 |
|
|
Bad oral hygiene |
43(63.2%) |
17(60.7%) |
0.817 |
|

Round 2
Reviewer 4 Report
Dear Editor,
The manuscript titled "Dental criteria could alert for malnutrition risks and inappropriate choice of food texture in older subjects with dementia: An analytical cross-sectional study" is revised as per the comments suggested. All the concerns have now been satisfactorily responded to. Thus, the manuscript can be accepted for publication.